# Coordination of RNA Processing Regulation by Signal Transduction Pathways

**DOI:** 10.3390/biom11101475

**Published:** 2021-10-07

**Authors:** Veronica Ruta, Vittoria Pagliarini, Claudio Sette

**Affiliations:** 1Department of Neuroscience, Section of Human Anatomy, Catholic University of the Sacred Heart, 00168 Rome, Italy; veronica.ruta-collaboratore@unicatt.it (V.R.); vittoria.pagliarini@unicatt.it (V.P.); 2Organoids Facility, IRCCS Fondazione Policlinico Universitario Agostino Gemelli, 00168 Rome, Italy; 3Laboratory of Neuroembryology, IRCCS Fondazione Santa Lucia, 00143 Rome, Italy

**Keywords:** signal transduction, alternative splicing, alternative polyadenylation, stress response

## Abstract

Signal transduction pathways transmit the information received from external and internal cues and generate a response that allows the cell to adapt to changes in the surrounding environment. Signaling pathways trigger rapid responses by changing the activity or localization of existing molecules, as well as long-term responses that require the activation of gene expression programs. All steps involved in the regulation of gene expression, from transcription to processing and utilization of new transcripts, are modulated by multiple signal transduction pathways. This review provides a broad overview of the post-translational regulation of factors involved in RNA processing events by signal transduction pathways, with particular focus on the regulation of pre-mRNA splicing, cleavage and polyadenylation. The effects of several post-translational modifications (i.e., sumoylation, ubiquitination, methylation, acetylation and phosphorylation) on the expression, subcellular localization, stability and affinity for RNA and protein partners of many RNA-binding proteins are highlighted. Moreover, examples of how some of the most common signal transduction pathways can modulate biological processes through changes in RNA processing regulation are illustrated. Lastly, we discuss challenges and opportunities of therapeutic approaches that correct RNA processing defects and target signaling molecules.

## 1. Introduction

Signal transduction pathways coordinate most cellular functions and represent a fast and tunable way for the cell to adapt to changes in the surrounding environment. Signaling pathways can be propagated by second messenger molecules, such as cyclic nucleotides (i.e., cAMP and cGMP), that diffuse in the cell and allosterically modulate the activity of proteins, or by direct post-translational modification (PTM) of the target proteins by dedicated enzymes. The most common and studied type of PTM is protein phosphorylation, a reversible covalent addition of a phosphate group to serine, threonine or tyrosine residues that is catalyzed by protein kinases and erased by phosphatase enzymes [1]. However, other amino acids can be reversibly modified post-translationally, including acetylation and methylation of arginine and lysine residues, ubiquitination and sumoylation of lysine residues, S-palmitoylation of cysteine, serine and threonine residues, or glycosylation of asparagine, serine and threonine residues [2,3]. 

Signal transduction pathways organize basically all processes in the cell and are endowed with several checkpoint and feedback mechanisms that fine tune the extent of their activation and therefore, the response of the cell. One of the main responses of the cell to a change in the surrounding environment is modulation of the expression of genes that encode proteins and RNAs involved in the specific biological processes that need to be activated or repressed. Gene expression changes underlie cell proliferation, cell differentiation or programmed cell death when insults have caused extensive damage. Activation of the expression of a specific gene requires binding of the transcriptional machinery to its promoter region and this event is favored by transcription factors and chromatin remodeling proteins. These factors are recruited to sequence-specific elements within the promoter or in the flanking regions and make the chromatin more accessible to the transcriptional machinery. Notably, most, if not all, transcription factors are subjected to PTM, which modifies their activity and/or interaction with other factors or with the RNA polymerase II (RNAPII) enzyme [4,5,6]. Likewise, PTM of histones is crucial to determine the accessibility of their chromatin to the transcriptional apparatus and is a main determinant of gene expression regulation in response to signaling pathways activated by external or internal cues [7,8,9]. Moreover, once transcription has started, nascent transcripts need to be processed in order to remove the non-coding intronic sequences and to modify the 5′ and 3′ ends to preserve the integrity of the mature transcript from exonuclease-mediated degradation. In particular, two of these processes—the splicing of introns and the cleavage and polyadenylation of the 3′ end of the transcript—are extensively modulated by signaling pathways through PTM of the factors involved in these mechanisms. In this review, we will briefly describe the main features of the splicing and polyadenylation processes and illustrate selected examples of how signal transduction pathways impinge on their regulation in eukaryotic cells.

## 2. Regulation of Nuclear RNA Processing

### 2.1. The Spliceosome and the Splicing Reaction

In eukaryotic cells, nascent transcripts comprise coding sequences—the exons—separated by non-coding sequences—the introns. These precursor mRNAs (pre-mRNAs) are not functional for protein synthesis until the introns are removed and the exons are joined. The spliceosome is the highly dynamic macromolecular machinery that recognizes the exon–intron boundaries and operates the splicing of the intronic sequences from the pre-mRNA [10]. The spliceosome is formed by five small nuclear ribonucleoprotein particles (snRNPs) named U1, U2, U4, U5 and U6 and other core and auxiliary proteins that dynamically associate with the snRNPs. Each snRNP comprises a small U-rich nuclear RNA (snRNA) and several proteins that are required for the function of the snRNP and catalysis of splicing [10]. 

In the initial step of splicing, the U1 snRNP is recruited at the 5′ splice site, while the 3′ splice site is recognized by non-snRNP factors known as splicing factor 1 (SF1) and the complex formed by the U2 auxiliary factors of 35 (U2AF35) and 65 (U2AF65) kilodaltons. SF1 recognizes the invariant A nucleotide, named the branch point, which serves as the docking site for the first trans-esterification reaction; U2AF35 binds the invariant AG dinucleotide at the 3′-end of the intron; U2AF65 is recruited to a polypyrimidine-rich sequence localized between the branchpoint and the 3′ splice site (Figure 1A) [10,11]. Following these protein–RNA interactions, U2 is recruited to the branch point in the pre-mRNA to form the A complex, or pre-spliceosome. Complex B is then formed by the recruitment of the tri-snRNP U4/U6 and U5. After conformational rearrangements and the dissociation of the U1 and U4 snRNPs, the spliceosome enters in its activated form, named B active complex, and catalyzes the two trans-esterification reactions required for intron excision and exon joining (Figure 1A) [10,11]. Several of the proteins that are associated with the snRNPs play a key role in these processes. For instance, the pre-mRNA-processing-splicing factor 8 (Prp8) of the U5 snRNP forms a scaffold that stabilizes the RNA catalytic core and maintains an open arrangement for the intron substrate, whereas the Brr2 helicase promotes the unwinding of the U4/U6 duplex. Other proteins, like Prp24, Prp3 and Prp4, are instead required for U4/U6 and U4/U6–U5 formation [10,11]. At the end of the splicing reaction, the spliceosome is disassembled and its proteins are recycled for new splicing processes.

### 2.2. The Cleavage and Polyadenylation Complex and 3′ End Processing of Transcripts

During transcription, the RNAPII does not autonomously stop at the end of the transcription unit to terminate the transcript. Indeed, the definition of the end of a pre-mRNA requires the cleavage of the transcript and the subsequent modification of the free 3′-end by addition of a non-templated poly(A) tail [12]. The polyadenylation process not only protects the mRNA from enzymatic degradation, but also promotes transcription termination and favors the export and translation of mRNA in the cytoplasm [13,14]. The cleavage and polyadenylation site (pA) is generally defined by a polyadenylation signal (PAS). Most human pAs comprise a canonical PAS sequence (AAUAAA) localized 10–40 nucleotides upstream of the cleavage site. The PAS recruits the cleavage and polyadenylation specificity factor (CPSF) complex and flags the pA for 3′-end processing [14], while contributing to slow down the RNAPII and favoring transcription termination [12]. Moreover, additional cis-acting RNA sequences flanking the pA, named regulatory upstream (USE) and downstream (DSE) sequence elements [14], recruit other trans-acting factors that cooperate with the CPSF complex to execute the cleavage and polyadenylation (C/P) reactions. In mammals, USEs include the UGUA and U-rich motifs, whereas DSEs comprise U- and GU-rich motifs [14]. Binding of the cleavage stimulation factor (CSTF) complex to DSE helps the CPSF to define the pA [14,15]. In addition, recruitment of cleavage factor I (CFIm) and CFIIm sub-complexes to USEs and DSEs, respectively, contributes to selection of the pA [14,15]. After cleavage, the free 3′-end is polyadenylated by the poly(A) polymerase (PAP) enzyme and the RNA downstream of the cleavage site is degraded by an exonuclease, whose activity facilitates transcription termination and the release of the RNAPII from the DNA template [12].

### 2.3. Alternative Splicing and Alternative Polyadenylation: Evolutionary Devices That Amplify Genome Complexity and Plasticity

Splicing of introns is an essential step in the processing of pre-mRNAs and impaired execution of this reaction is lethal. However, although the first requirement for efficient and accurate splicing is the recognition of the exon–intron junctions, these boundaries are not marked by highly conserved sequence elements. Indeed, beside the almost invariable dinucleotides at the 5′ (GT) and 3′ (AG) ends of introns, the sequences defining the exon–intron junctions are highly degenerate in higher eukaryotes [16]. To help the spliceosome to identify the correct splice sites, additional sequence elements, generally defined as splicing enhancers and silencers, are present in both exons and introns. These elements are recognized by sequence-specific RNA-binding proteins (RBPs), which act as trans-acting splicing factors that bind the pre-mRNA and regulate splicing decisions by either promoting or inhibiting the positioning of the spliceosome at the splice sites (Figure 1B) [16,17]. The most characterized families of regulatory splicing factors are the serine–arginine rich proteins (SR) and the heterogeneous nuclear ribonucleoproteins (hnRNPs), which often perform opposite actions in a concentration dependent manner [18]. The interplay between antagonistic splicing factors can determine whether an exon is included or not in the mature mRNA through a process named alternative splicing (AS; Figure 1B) [16]. The advent of high-throughput RNA sequencing technologies has now revealed that virtually all multi-exon human genes undergo AS regulation [16,19], thus yielding multiple transcript variants that often encode proteins with different, or even opposite, functions [19,20]. The flexible nature of AS allows highly dynamic regulation of gene products and is susceptible to changes in the external and internal cellular environment, thus fine-tuning gene expression in response to the needs of cells. Likewise, most genes in higher eukaryotes comprise multiple pAs and can undergo alternative polyadenylation (APA) [14]. APA can lead to either changes in the protein-coding sequence of transcripts (CDS-APA) or in the length of their regulatory 3’ untranslated region (UTR-APA) [14,15,21]. CDS-APA are defined by the presence of at least one alternative pA within the transcription unit (internal pA, IPA), either in an exon that is alternatively spliced or in introns that are spliced inefficiently and whose splice sites enter into competition with the intronic pA [22,23]. In UTR-APA, instead, the alternative pAs are localized in the last exon downstream of the stop codon and their differential selection affects the length of the UTR and its regulatory potential [14,15,21]. 

Notably, AS and APA are present in all eukaryotes, from yeast to humans, but their contribution to transcriptome diversity increases with organism complexity and has contributed to amplifying the coding potential of eukaryotic genomes without the need of increasing the number of genes [14,15,16,24]. Several mechanisms contribute to the regulation of these RNA processing mechanisms, including transcription dynamics and epigenetic modifications of DNA and histones [25,26]. However, PTMs of core and auxiliary proteins of the splicing and C/P machineries are the most common determinants of splicing regulation. Several signal transduction pathways have been shown to regulate key developmental or differentiation processes by modifying the expression or activity of factors involved in both AS and APA through various types of PTMs. In the following paragraphs, we will summarize key examples of how specific PTMs impact RNA processing events in eukaryotic cells.

## 3. Post-Translational Modifications and RNA Processing

### 3.1. Sumoylation

SUMO proteins are small, ubiquitin-like molecules that are covalently attached to target proteins and are evolutionary conserved from protozoa to metazoa, including plants and fungi. Conjugation of a SUMO molecule to a target protein (sumoylation) is a four-step process resulting in a stable connection between a lysine residue in the substrate and the activated SUMO protein catalyzed by the UBC9 conjugation enzyme. The SUMO modification is reversible and can be removed by specific proteases named SENPs and by the deSUMOylating isopeptidase1/2 (DESI1/2) and ubiquitin-specific protease-like 1 (USPL1) [27,28]; this PTM generally stabilizes the target protein and regulates protein–protein interactions as well as protein localization and function [27,28].

SUMOs can be attached as single molecules or in chains, with several molecules being linked to the same residue. Five SUMO isoforms have been reported in humans, which share different grades of sequence identity and are involved in different functions. SUMO1 is not able to form chains and its conjugation can modify the activity and function of the target proteins [29]. SUMO2 and 3 have high degrees of amino acid identity and modify stress proteins. SUMO4 has been implicated in the regulation of protein stability and localization [27,28]. Lastly, SUMO5 is not expressed in mice and is specific to particular cells and tissues, such as blood cells and testis [30]. 

Many RBPs are substrates for sumoylation or comprise SUMO-interacting motifs (SIMs) [31,32]. In 2007, Vethantham and colleagues identified two key regulators of the C/P process, CPSF73 and symplekin, as targets of SUMO2/3 [33]. By using sumoylation-deficient mutant cells, it was shown that this PTM is essential for the function of symplekin. In addition, inhibition of sumoylation by depletion of UBC9 or overexpression of a SUMO protease impaired the assembly and activity of specific 3′-end complexes [33]. Sumoylation is also involved in splicing catalysis. Initial studies reported that the SUMO E3 ligase PIAS1 co-purifies with the spliceosome [34]. Moreover, components of the SUMO pathway co-localize with splicing factors in Cajal bodies [35,36]. More recently, a clear role of sumoylation in the splicing process has been demonstrated by performing in vitro splicing assays with HeLa nuclear extract [37]. Under these in vitro conditions, the presence of exogenous recombinant SENP1 protease reduced splicing efficiency, whereas addition of SUMO-activating and conjugating enzymes rescued the defect. Furthermore, it was demonstrated that the SR protein SRSF1 regulates sumoylation of key core and auxiliary spliceosomal proteins, such as U2AF65, Snu114, Prp28 and Prp3 [37,38]. Interestingly, these studies indicated that while a sumoylation-deficient Prp3 mutant was still able to associate with U4/U6 snRNAs, its ability to interact with the U2 and U5 snRNPs was impaired, thus reducing the recruitment of the active spliceosome and compromising splicing efficiency [37,38]. These observations support a direct role of protein sumoylation in multiple steps of RNA processing regulation.

### 3.2. Ubiquitination

Ubiquitination is a dynamic modification, by which the C-terminal glycine of a ubiquitin molecule forms an isopeptide bond with the amino group on the side chain of lysine residues of the target protein (or other ubiquitin-forming ubiquitin chains). Alternatively, it can also form a peptide bond between its C-terminal glycine and the N-terminal protein of a target [39]. The process of ubiquitination involves three classes of enzymes—E1, E2 and E3—that respectively activate, conjugate and attach ubiquitin to the target protein. While the canonical role of ubiquitination involves targeting proteins to the proteasome for degradation, this PTM can be also used as a regulatory signal modulated by the balance between the activity of E3 ubiquitin ligases and deubiquitinating enzymes [40]. Specifically, as there are seven lysine residues in ubiquitin (K6, K11, K27, K29, K33, K48 and K63) [41], linkages between the C-terminal glycine and one of these seven lysine residues on a different ubiquitin molecule results in functionally distinct types of chains. The resulting chain influences the downstream pathway that is activated. For instance, linkage to K11 or K48 results in the degradation of the targeted protein [42,43,44], whereas linkage to K63 results in the regulation of protein localization, assembly of DNA repair complexes or transcriptional activation [45].

The proteins regulated by ubiquitination generally contain ubiquitin-binding domains (UBDs) that promote their recognition [41]. Interestingly, non-canonical UBDs have been identified in spliceosomal proteins, like the Jab1/MPN domain of Prp8 [46], an essential U5 snRNP component. The correlation between ubiquitination and splicing was first reported in yeast, where deletion of the hub1 gene, encoding a ubiquitin-like protein, resulted in splicing defects [47]. Moreover, inhibition of ubiquitin recognition or removal of ubiquitin conjugates was shown to accelerate the unwinding of the U4/U6 snRNPs from U5 snRNP, thus suggesting that ubiquitination is important to stabilize the triple snRNP (U4/U6–U5) conformation [48]. This study also identified Prp8 as a ubiquitin conjugate that suppresses Brr2-catalyzed disassembly of the spliceosome. A model was proposed in which inhibition of Brr2 by the ubiquitinated Prp8 leads to spliceosome assembly and stabilization of the triple snRNP (U4/U6–U5). Following de-ubiquitination of Prp8, Brr2 is free to promote unwinding of U4/U6, resulting in activation of the spliceosome [48].

Shortly afterwards, it was also demonstrated that the E3 ligase named Prp19 increases the affinity between the ubiquitinated Prp3 protein and Prp8, thus contributing to stabilization of the triple snRNP. On the other hand, Usp4 and Sart3 promote de-ubiquitination and recycling of Prp3, and this modification weakens its interaction with Prp8 and allows for the dissociation of U4 during activation of the spliceosome [49]. A subsequent screen for other spliceosomal proteins regulated by the ubiquitination/de-ubiquitination cycle during mitotic progression identified the U4 snRNP component Prp31 as a target of Prp19 E3 ligase activity. Ubiquitination of Prp31 increases its affinity for Prp8 and this PTM is reversed by USP15. Moreover, it was reported that Sart3, Usp4 and Usp15 form a complex in order to de-ubiquitinate Prp3 and Prp31 simultaneously. Collectively, these studies suggested that ubiquitination plays a key regulatory role in the rearrangements of the spliceosome [50].

### 3.3. Methylation

Arginine methylation is a widespread PTM involved in signal transduction, DNA damage repair, gene transcription, splicing regulation and RNA metabolism. Arginine methylation is catalyzed by a family of nine arginine methyltransferases (PRMT1-9) [51,52], which produce three types of modifications: ω-NG-mono-methyl-arginine (MMA), ω-NG,NG-asymmetric di-methyl-arginine (aDMA) and ω-NG, N’G-symmetric di-methyl-arginine (sDMA). On the basis of the final product they catalyze, PRMTs are divided into three types [52]: Type I: PRMT 1, 2, 3, 4, 6, and 8 are involved in the formation of MMA and aDMA; Type II: PRMT 5 and 9 are involved in the formation of MMA and sDMA; Type III: PRMT7 is involved in the formation of MMA.

Arginine methylation is involved in the maturation of the spliceosomal snRNPs. Assembly of snRNPs require association of the Sm proteins with newly synthesized U1, U2, U4 and U5 snRNAs. Once the snRNPs are formed in the cytoplasm, they are imported into the nucleus, where they can regulate pre-mRNA splicing [53]. Symmetric dimethylation of Sm/LSm proteins, like SmD1, SmD3, SmB/B’ and LSm4, by PRMT5 is important for recognition by the survival motor neuron protein (SMN) during the assembly of the snRNP core particles [54]. Since this functional pathway is crucial to yield the spliceosomal snRNPs, its inhibition affects spliceosome formation and pre-mRNA splicing [55]. Accordingly, knockdown of PRMT5 activity was later shown to cause widespread defects in splicing in eukaryotic cells [56]. Likewise, loss of PRMT5 in other organisms, like *Arabidopsis thaliana* (AtPRMT5) [57] and *Drosophila melanogaster* (Dart5) [58], causes defects in pre-mRNA splicing. Moreover, in the yeast *Saccharomyces cerevisiae*, co-transcriptional recruitment of splicing factors was shown to be promoted by arginine methylation. Yeast mutants lacking the methyltransferase Hmt1, or defective for its catalytic activity, display aberrant recruitment of splicing factors on their target transcripts. In particular, arginine methylation was shown to regulate the association between the U1 snRNP component Snp1 and the SR-like protein Npl3 [59].

Besides regulating the splicing machinery directly, arginine methylation was also shown to affect RNA metabolism by modulating the localization of the proteins involved. A high-throughput proteomic approach identified over 200 arginine-methylated proteins and revealed that this PTM affects several RBPs, suggesting a widespread regulation of RNA metabolism by this type of PTM [60]. In particular, methylation of hnRNP and SR proteins was shown to affect their cellular localization [61,62,63]. For instance, mutation of the methylated arginine residue in SRSF1 (previously known as SF2/ASF) mislocalizes this SR protein in the cytoplasm [63]. On the other hand, hnRNPs are methylated at the arginine of the C-terminal arginine–glycine–glycine-rich (RGG) repeat [64], which can also contribute to RNA binding together with the prototypical RNA recognition motif (RRM) at their N-terminus. Thus, arginine methylation appears as a PTM involved in multiple key steps of RNA metabolism, from the functional assembly of core factors to fine-tuned modulation of localization and function of specific RBPs.

### 3.4. Acetylation

Lysine acetylation is important for the regulation of many cellular processes. This reversible PTM can compete with methylation, ubiquitination and sumoylation for regulation of the stability, localization and interactions of the target proteins [65,66]. The balance between histone acetylation and deacetylation needs to be accurately controlled to ensure proper gene expression and both excessive hyperacetylation, as a consequence of deletion/inhibition of histone deacetylate enzymes (HDACs), or hypoacetylation, as consequence of deletion/inhibition of histone acetyltransferase (HAT) activity, is deleterious for the cell [67,68]. The general control non-repressed 5 protein (GCN5) was the first HAT reported to link histone acetylation with transcriptional activation [69]. GCN5 was also found to promote the recruitment of the U2 snRNP to pre-mRNAs [70] and deletion of the Gcn5 gene, or inhibition of GCN5 catalytic activity, is synthetic lethal with deletion of genes encoding U2 snRNP components (i.e., Msl1 or Lea1) [70]. GCN5 is recruited to intronic sequences of genes and although GCN5-mediated acetylation is mainly observed at promoters, deletion of HDAC activity results in H3K9/K14 acetylation in the body of the genes as well. This finding suggests that acetylation within the gene body is hidden by the rapid action of HDACs but may play a transient role during transcription. Indeed, in the absence of HDAC activity, the interaction between the U2 snRNP and the branchpoint region is persistent and recruitment of downstream snRNPs is delayed, thus suggesting a mechanism in which histone acetylation and spliceosome dynamics are coupled [71]. In line with this hypothesis, histone acetylation marks were shown to contribute to the recruitment and rearrangement of splicing factors during co-transcriptional splicing [71]. Moreover, SF3B3/SAP130, a component of the SF3B complex of the U2 snRNP, was shown to interact with the mammalian SAGA-like HAT complex (STAGA) in mammalian cells, further linking spliceosome assembly and histone acetylation [72]. Accordingly, inhibitors of HATs and HDACs were shown to block spliceosome assembly in vitro at different steps, confirming the key role of acetylation in the regulation of the splicing process [73].

Other evidence connects protein acetylation more directly with RNA processing regulation. For instance, acetylation of the signal transduction and activation of RNA (STAR) protein Sam68, which is involved in AS and APA [23,74,75], was shown to enhance its affinity for target RNA sequences; this PTM was increased in cancer cells [76]. More recently, it was shown that the HAT known as CREB-binding protein (CBP) is directly involved in snRNP biogenesis by promoting K119 acetylation of SMN in vivo and that the SMN acetylation status controls its subcellular localization and regulates its binding with proteins required for snRNP biogenesis [77].

### 3.5. Phosphorylation

Phosphorylation is the most extensively studied PTM and is basically involved in all cellular processes in both healthy and disease conditions. For instance, dysregulation of phosphorylation pathways is often a trigger of pathologies, such as human cancers, and many typical oncogenes, such as HER2 and KRAS, directly or indirectly activate downstream phosphorylation cascades. Threonine, serine and tyrosine residues are the common targets of phosphorylation by protein kinases in mammalian cells, whereas reversion of this PTM is catalyzed by phosphatases. In line with the crucial role played by protein phosphorylation, protein kinases represent the largest gene family in the human genome, comprising ~2–5% of total genes [78]. Phosphorylation affects conformational changes in the target protein by modifying its affinity for other proteins, activating/deactivating its function, changing the subcellular localization or facilitating its degradation by the ubiquitin–proteasome complex [78].

Cycles of phosphorylation/dephosphorylation of many spliceosomal proteins are crucial for the correct execution of the catalytic steps of the splicing reaction. In particular, dephosphorylation by the protein phosphatases PP1 and PP2A fine-tunes the splicing process and is essential for its outcome. In an elegant paper, Shi and colleagues reported that depletion of PP1 and PP2A affects splicing and impairs the accumulation of the final splicing products. PP1/PP2A depletion does not affect the formation of the spliceosome complexes A, B and C; however, the C complex accumulates in a stalled conformation, resulting in defects in the second step of the splicing reaction [79]. Moreover, they proved that SAP155 and U5-116 kDa, components of the U2 and U5 snRNPs, respectively, are substrates of PP1 and PP2A, thus highlighting the key role played by the protein phosphorylation–dephosphorylation cycle in the second step of splicing [79].

A few years later, SR protein kinase 2 (SRPK2) was identified as a regulator of the phosphorylation of the spliceosomal protein PRP28 [80]. SRPK2 associated with snRNPs and its knockdown affected spliceosomal B complex formation and cell viability. The snRNP protein PRP28 is a direct substrate of SRPK2 and depletion of this kinase led to PRP28 hypophosphorylation in the cell. Further investigation also demonstrated that PRP28 phosphorylation is essential for stable association with the tri-snRNP complex and for the assembly of the spliceosome B complex [80].

As their name suggests, SRPKs are mainly known for their ability to phosphorylate SR proteins. They generally catalyze this reaction in the cytoplasm, whereas another class of kinases, the CDC-like kinase (CLK1-4) family, promotes phosphorylation of these splicing factors in the nucleus [81,82]. Moreover, most RBPs involved in the splicing and polyadenylation processes have been shown to be phosphorylated, with functional consequences of this type of PTM on their subcellular localization, function and/or fate in the cell. Since an extensive description of the role of several protein kinases on the regulation of RBP function and RNA processing has been already reported [81,82], in this review we will focus on specific pathways that impinge on the regulation of RNA processing events by PTM of RBPs.

## 4. Post-Translational Modification of RNA-Binding Proteins

RBPs govern the maturation and function of the target RNAs, thus regulating gene expression at all steps of the RNA processing/maturation cycle including pre-mRNA transcription, splicing, cleavage and polyadenylation in the nucleus and mRNA stability, localization, editing and translation in the cytoplasm [83,84].

The activity of RBPs is finely modulated by signal transduction pathways that promote various types of PTMs, which contribute to the dynamic nature of RBP function and can also spatially organize RBPs within the cell. A complete list of PTMs described for all identified RBPs is beyond the scope of this review. The selected examples reported here are intended to illustrate the intricate mechanisms underlying the post-translational regulation of RBPs and its impact on key biological processes.

### 4.1. PTMs and Subcellular Localization of RBPs

PTMs regulate the subcellular localization of many RBPs, either by affecting their intranuclear and intracytoplasmic localization or their nucleocytoplasmic shuttling. The STAR protein Sam68 represents an excellent example in this sense [85]. Phosphorylation of tyrosine residues in the carboxyl-terminal domain of Sam68 by SRC-family kinase FYN and the related breast tumor kinase BRK caused its accumulation in nuclear granules [75,86], whereas mutation of a single tyrosine in the nuclear localization signal to the non-phosphorylatable phenylalanine completely blocked nuclear localization of Sam68 [86]. Interestingly, tyrosine phosphorylation of Sam68 negatively affects its interaction with RNA and with the RBP hnRNP A1, thus impairing its ability to modulate splicing of target pre-mRNAs [75,87,88]. Another RBP whose cellular localization is tightly regulated by PTMs is the fused in sarcoma/translated in liposarcoma (FUS) protein, which is mutated in a familial form of amyotrophic lateral sclerosis (ALS). Arginine methylation of FUS by PRMT1 controls its nucleocytoplasmic shuttling and this balance is altered in ALS-related mutants, with consequent accumulation of FUS in cytoplasmic aggregates that are typical of the disease [89]. Likewise, acetylation of FUS at lysine 510 by the CBP/p300 protein impeded its nuclear import via transportin 1 (TRN1) and sequestered this RBP in cytoplasmic stress granule-like inclusions [90]. The subcellular localization of TDP43, another ALS-related RBP, was also regulated by PTM; sumoylation of TDP43 promoted its nuclear localization and splicing activity. Conversely, inhibition of this PTM caused accumulation of TDP-43 in cytoplasmic stress granules [91], suggesting that dysregulation of TDP-43 sumoylation may contribute to the toxic aggregates that are a prototypic feature of ALS. 

The widespread impact of phosphorylation on the localization and activity of many members of the two main families of splicing factors, the SR and hnRNP proteins, has been extensively reviewed elsewhere [81,82,92]. Here, we provide a few explanatory examples of how the localization of these RBPs is modulated by PTMs. The subcellular localization of SR proteins is mostly regulated by kinases of the SRPK and CLK families [81]. These two classes of enzymes preferentially phosphorylate SR proteins on different residues and cooperate to regulate the localization of SR proteins. For instance, both CLK- and SRPK-dependent phosphorylation of SRSF1, a prototypic SR protein, contribute to relocalize this splicing factor from nuclear speckles to the nucleoplasm [93,94,95,96]. Upon translocation into the nucleus, SRPK1 forms a binary complex with CLK1, displacing the interaction of this latter kinase with SRSF1 and favoring the recruitment of SRSF1 to its RNA targets [93,94]. Moreover, phosphorylation of SR proteins was also reported to affect their nuclear import by modulating their interaction with SR-specific transportin proteins [95,96]. Likewise, the subcellular localization of several hnRNPs is subjected to regulation by PTMs [81]. A clear example is provided by hnRNP K, whose modular structure permits it to play a crucial role in many distinct biological processes [97]. Phosphorylation of hnRNP K by extracellular signal-regulated kinase (ERK) at serine S284 and S353 affects its nuclear–cytoplasmic trafficking and its ability to regulate protein translation [98]. Conversely, arginine methylation of its RGG motif (a.a. 280–307) by PRMT1 strengthens the nuclear retention of hnRNP K [99]. Indeed, nuclear localization was significantly impaired in mutant hnRNP K lacking the PRMT1 methylation region or upon pharmacological inhibition of methylation [99]. Moreover, hnRNP K glycosylation mediated by O-linked N-acetylglucosaminyltransferase is also implicated in its nuclear translocation in cholangiocarcinoma cells [100].

Removal of PTMs in response to specific stress stimuli may also be determinant for the function and activity of some RBPs. In the case of hnRNP A2B1, herpes simplex virus 1 (HSV-1) infection induced demethylation at Arg226 by the demethylase JMJD6, which promoted its translocation from nucleus to cytoplasm. In the cytoplasm, hnRNP A2B1 activated the TANK-binding kinase 1-interferon regulatory factor 3 (TBK1-IRF3) pathway and induced IFN-α/β production, thus amplifying the innate immune response to DNA viruses [101]. Importantly, mutation of arginine 226 to alanine (R226A) within its RGG domain was sufficient to enhance hnRNP A2B1-mediated Ifnb1 expression [101], highlighting how reversible arginine methylation controls a key cellular process in response to an external stimulus by modulating the subcellular localization and function of an RBP.

Arginine methylation controls the subcellular localization of several other RBPs. Although in some cases the mechanism varies or is unknown, this PTM generally promotes the retention of RBPs into the nucleus. For instance, this effect was also reported for hnRNP Q [61], the nuclear poly(A) binding protein (PABPN1), an RBP that intervenes in the final stages of RNA maturation [102], SERBP1 [103] and Sam68 [104]. While for hnRNP Q the mechanism underlying arginine methylation-mediated nuclear retention was not investigated [61], arginine methylation of PABPN1 was shown to reduce its affinity for the transportin involved in its shuttling to the cytoplasm [102]. Conversely, arginine methylation of SERBP1 guarantees its cytoplasmic localization [103]. In the case of Sam68, its arginine methylation relies on its functional interaction with PRMT1 and, as for hnRNP Q and PABPN1, it promotes nuclear accumulation of the RBP [104].

Other less known PTMs have been also proposed to modulate the intracellular localization of RBPs. For instance, Poly(ADP)-ribosylation of hnRNP A1 is required for its cytoplasmic translocation and recruitment in stress granules upon phase separation [105]. Furthermore, Poly(ADP)-myristoylation was shown to regulate the distribution of the neuronal fragile X-related protein 2 (FXR2P) to proximal axon segments [106]. Collectively, these non-exhaustive examples indicate the strong impact of multiple PTM types on the subcellular localization of RBPs, with consequent modulation of their multiple functions both in the nucleus and in the cytoplasm. 

### 4.2. PTMs and Activity of RBPs

PTMs can affect RBP activity by modulating their interaction with specific RNAs and/or proteins, their stability or by influencing the deposition of other PTMs in the same RBP. Here we illustrate recent examples that have contributed in the last decade to clarifying molecular mechanisms underlying the PTM-dependent regulation of RBP activities.

The Hu antigen R (HuR) protein, also known as embryonic lethal, abnormal vision, drosophila homolog-like 1 (ELAV1), is a ubiquitous RBP belonging to the family of Hu proteins, which also comprises three other members (HuB, HuC, HuD) expressed exclusively in neurons. HuR is predominantly localized in the nucleus. However, upon exposure to stressful conditions or mitogens, HuR translocates to the cytoplasm, where it stabilizes target mRNAs and/or modulates their translation [107,108]. Nucleocytoplasmic shuttling of HuR is modulated by phosphorylation in its central ‘hinge” region and is mediated by several kinases, including protein kinase C (PKC), cyclin-dependent kinase 1 (CDK1) and checkpoint kinase 2 (CHK2). Binding of HuR to its targets is finely regulated by phosphorylation; for instance, phosphorylation of serine 242 in the hinge domain of HuR, presumably by CDK1, impaired its cytoplasmic localization, preventing its binding to cyclin A2 and cyclin B1 transcripts and reducing their translation [109]. Interestingly, overexpression of a non-phosphorylatable HuR mutant, in which serine 242 was substituted with alanine (S242A), increased cyclin A2 and B1 expression and positively regulated cell proliferation [109]. Moreover, under different stresses, phosphorylation of HuR by CHK2 caused its dissociation from target mRNAs, thus promoting their decay and reducing their expression. For instance, following H_2_O_2_ exposure and consequent oxidative stress, CHK2 is activated and phosphorylates HuR at serine 88 and 100 and at threonine 118. Site-directed mutagenesis experiments showed that CHK2-mediated phosphorylation of serine 100 is responsible for its dissociation from the 3′UTR of the longevity and stress-response protein SIRT1, thus destabilizing it [110]. The consequent decay of SIRT1 mRNA and reduced expression of SIRT1 protein impaired cell survival in response to stress [110]. Likewise, occludin mRNA, encoding a transmembrane tight junction protein that plays an important role in the assembly and regulation of the epithelial barrier, is also negatively regulated by CHK2-dependent phosphorylation of HuR [111]. However, CHK2-mediated phosphorylation was also reported to increase the affinity of HuR for other target mRNAs. Indeed, serine 100 phosphorylation of HuR increased its binding to the 3′UTR of the MYC mRNA and promoted its translation. It is at present unclear how phosphorylation in the same residue (i.e., CHK2-dependent phosphorylation of serine 100) may both reduce (SIRT1) and increase (MYC) the binding affinity of HuR for a specific mRNA. It is possible that the secondary structures in the 3′UTR of the transcript and/or its proximity with the binding of additional regulatory factors in different mRNAs play a significant role in determining HuR binding strength.

Another RBP whose activity is finely regulated by several modifications is Sam68 [85]. Sam68 is a multifunctional protein with documented roles in signal transduction pathways, where it generally acts as scaffold for the recruitment of other proteins and in various steps of RNA metabolism, from transcription to splicing and translation of selected transcripts [112]. Tyrosine phosphorylation of the C-terminal region of Sam68 by SRC-family kinases, such as SRC, FYN and BRK, interferes with the ability of Sam68 to interact with proteins containing Src Homology 3 (SH3) domains and with RNA [112,113], thus affecting many of its functions. For instance, Sam68-dependent splicing of the BCL-X and CCND1 transcripts is strongly suppressed by FYN-mediated phosphorylation [75,87]. On the other hand, serine/threonine phosphorylation of Sam68 by ERKs [74] or Ca2+-calmodulin kinase IV (CAMKIV) [114] promoted its splicing activity, as well as the ability of Sam68 to induce translation of target transcripts in the cytoplasm [115]. Moreover, other PTMs, such as arginine methylation and acetylation, were also shown to affect Sam68 activity by modulating its binding affinity for proteins or RNA [76,85,104,116,117].

The function(s) of several RBPs is regulated by PTMs that modify their ability to interact with other proteins in the cell. PRMT1-mediated arginine methylation was shown to increase the interaction of hnRNP K with the transcription factor p53 in response to UV irradiation, thus regulating its transcriptional activity [118,119]. Conversely, this same PTM reduced the interaction of hnRNP K with the tyrosine kinase SRC and prevented its activation [120]. Since SRC-mediated phosphorylation of hnRNP K drives translational activation of repressed mRNAs [121], its methylation may rapidly switch hnRNP K function from a translational regulator in the cytoplasm to a transcriptional regulator in the nucleus. For instance, the reticulocyte-15-lipoxygenase (r15-LOX) transcript, encoding a key enzyme in erythroid cell differentiation, is repressed at early stages of erythroid differentiation. Upon activation of SRC by hnRNP K in mature reticulocytes, and consequent phosphorylation of hnRNP K, the r15-LOX mRNA is translated [122]. Thus, arginine methylation of hnRNP K could be required to prevent premature activation of SRC and to restrict the time-window for the translation of its target mRNAs [121].

Interestingly, PRMT1-dependent methylation of hnRNP K also interfered with the activity of another kinase: PKCδ [123]. In this case, PRMT1-dependent methylation of arginine 296 and 299 in hnRNP K interferes with PKCδ-dependent phosphorylation of hnRNPK at serine 302 and guarantees its anti-apoptotic role following etoposide-induced DNA damage [123]. Although the molecular mechanism/s underlying the inhibition of hnRNP K phosphorylation is not fully understood, it was speculated that methylation of arginine 296 and 299 might sterically hinder the accessibility of proximal serine 302 to the catalytic site of PKCδ. Furthermore, the proximity of methylation and phosphorylation sites in a highly conserved region of hnRNP K suggests that the crosstalk between these PTMs might have acquired important functional roles in the regulation of gene expression during evolution [123].

PTMs also play a key role in the turnover rate of RBPs. For instance, phosphorylation of HuR by the IKKα and PKCα kinases in response to inhibition of glycolysis promotes the interaction of this RBP with the ubiquitin E3 ligase β-TrCP1 and its degradation [124]. On the other hand, while ubiquitylation of hnRNP K by the ubiquitin E3 ligase HDM2/MDM2 triggers its proteasomal degradation [119], phosphorylation by ATM or sumoylation by the E3 ligase polycomb protein (Pc2) stabilize hnRNP K in response to genotoxic stress, thus allowing a robust p53-mediated transcriptional response to DNA damage [125,126]. These selected examples illustrate how different PTMs can fine-tune the activity of RBPs and, in turn, regulate specific gene expression programs in time and space according to the needs of the cell.

## 5. Signaling Pathways That Regulate RNA Processing Machinery

Signaling pathways are characterized by a set of proteins which receive information from external or internal cues and convey it within the cell to activate a proper response. In this way, signaling pathways allow eukaryotic organisms to rapidly adapt to sudden changes in the external or internal environment and to cope with the altered situation. The ultimate response, especially when signals are durable in time, is to change the gene expression program of the cell to better fit with the new situation. Indeed, signaling pathways impinge on all steps involved in gene expression, from transcription to maturation and translation of RNA transcripts. Regulation of the splicing and polyadenylation processes greatly contributes to expanding the adaptation potential of cells. In the recent past, many studies have highlighted how signaling pathways globally impact RNA processing regulation directly, by interacting with and influencing the expression or activity of specific RBPs, or indirectly, by acting on their regulators. Herein, we will describe some of the signaling pathways that have been shown to control RNA processing steps and how the discovered connections have uncovered new regulatory layers in gene expression programs.

### 5.1. The PI3K/AKT Pathway and RNA Processing Regulation

The phosphoinositide 3-kinase (PI3K) signaling pathway transduces the signal of several growth factors and cytokines [127]. The main player in the PI3K pathway is the serine/threonine kinase AKT (also called PKB). Through the phosphorylation of its many cytosolic and nuclear targets, AKT regulates a multitude of cellular processes, such as metabolism, proliferation and survival [127]. Among others, AKT, and more generally the PI3K pathway, have been also shown to affect RNA maturation processes through regulation of the activity of several splicing factors and RBPs [81,82].

The three AKT isoforms, called AKT 1, 2 and 3, share a very similar kinase domain and play non-completely redundant roles in mammals. Indeed, mouse knockout models expressing only one AKT isoform showed distinct developmental phenotypes [128]. In this regard, it was also demonstrated that AKT isoforms differentially regulate AS to promote proliferation and invasion in lung cancer [129]. To identify the putative substrates that underlie the functional differences between AKT1–3, the authors performed a phospho-proteomics screen in murine lung fibroblasts expressing only one of the three isoforms. Interestingly, these kinases showed both unique and overlapping substrate specificities, with proteins involved in RNA processing, like the splicing factors hnRNP M and SRRM1, being enriched among the substrates of AKT1 and AKT3 [129]. Moreover, AKT1 and AKT3, but not AKT2, phosphorylated the transcription factor IWS1 (interacts with SUPT6H, CTD assembly factor 1) at serine 720 and threonine 721, respectively, thus favoring its interaction with the histone H3K36 methyl transferase SETD2 and promoting the recruitment of the latter to the RNAPII elongation complex. This AKT-dependent interaction of IWS1 represents a key event in the regulation of AS events downstream of the signaling pathway activated by the human fibroblast growth factor receptor 2 (FGFR2) (Figure 2A) [129]. Notably, FGFR2 encodes for two alternatively spliced variants that differ in the inclusion of one of two mutually exclusive exons (IIIb and IIIc), which are selectively expressed in epithelial and mesenchymal cells, respectively. Histone marks on chromatin regions surrounding these alternative exons can favor the recruitment of specific chromatin-binding proteins which, in turn, recruit specific splicing factors. In particular, SETD2-dependent trimethylation of H3K36 (H3K36me3) in exon IIIb favors the recruitment of the chromatin-binding protein MRG15 and PTBP1 (hnRNP I), thus repressing the inclusion of exon IIIb in the mature transcript and leading to the expression of mesenchymal FGFR-2-exon IIIc variant (Figure 2A) [130]. Noteworthily, lung carcinomas, displaying increased AKT-dependent phosphorylation of IWS1, are characterized by FGFR-2 IIIc expression and more malignant features [129]. Since AKT was also shown to directly phosphorylate other histone modifiers, such as the H3K9/H3K27 methyl transferase EZH2 [131], it is conceivable that the frequent dysregulation of the PI3K/AKT pathway observed in cancer cells leads to more global changes in epigenetic regulation of splicing programs through altered genome-wide recruitment of RBPs.

The PI3K/AKT pathway was also shown to control AS by influencing the activity or expression of splicing factors. For instance, AKT-dependent phosphorylation of hnRNP L promoted its binding to exon 3 of the caspase 9 pre-mRNA, thus out-competing hnRNP U and inducing the expression of an antiapoptotic exon 3-skipped splice variant [132]. Furthermore, the PI3K/AKT pathway can modulate the function of SR proteins directly [133,134,135,136,137] or indirectly, by affecting the activity of their regulators, such as SRPKs or CLKs [138,139,140]. Growth factor-induced phosphorylation of SRSF1 and SRSF7 by AKT enhanced the ability of these splicing factors to promote the inclusion of the extra domain A (EDA) exon in fibronectin (FN) mRNA and the expression of a splice variant that contributes to tumor growth and invasion [136,137,141]. Likewise, AKT-mediated phosphorylation of SRSF5 regulates PKC*β* AS after insulin stimulation, promoting the PKC*β* II variant, which regulates glucose transport less efficiently [134,135]. On the other hand, an example of indirect regulation of splicing by the PI3K/AKT pathway is illustrated by the work of Zhou and colleagues [140]. Upon activation of EGF signaling, AKT interacts with SRPKs and induces their autophosphorylation and dissociation from the HSP70 chaperone, which normally holds these kinases in the cytoplasm. Nuclear translocation of SRPKs follows, guided by HSP90, where they phosphorylate SR proteins. In this way, AKT-dependent nuclear translocation of SRPKs resulted in a massive reprogramming of AS upon mitogenic stimulation [140]. 

Alternatively, the PI3K/AKT pathway can impact AS programs by regulating the expression of splicing factors. The PI3K/AKT pathway is constitutively activated in Ewing sarcoma, an aggressive tumor of bone and soft tissues. Treatment of Ewing sarcoma cells with a dual inhibitor (BEZ235) of PI3K and the mechanistic target of rapamycin (mTOR), a downstream kinase in the pathway, caused extensive reprogramming of the cellular transcriptome, with thousands of genes modulated at either expression or splicing level [142]. Gene ontology analysis revealed “spliceosome” as one of the most enriched functional categories among the genes regulated by BEZ235 treatment, suggesting a specific activation of splicing as a feedback response of Ewing sarcoma cells to PI3K/AKT inhibition. In particular, the expression of hnRNP M was strongly upregulated. Moreover, treatment with BEZ235 induced the recruitment of hnRNP M to the splicing machinery and enhanced its effect on the AS of a large set of genes in response to inhibition of the PI3K/AKT pathway [142]. Similarly, expression of the RNA-binding motif 20 protein (RBM20) is also finely regulated by the PI3K/AKT pathway [143]. Interestingly, RBM20 levels were differently regulated by components of this pathway that act downstream of mTOR in neonatal rat cardiomyocytes stimulated with insulin. Knockdown of p70S6K1 reduced the expression of RBM20, whereas depletion of the negative regulator 4E-BP1 induced it [143]. Such PI3K/AKT/mTOR-dependent regulation acquires functional relevance due to the well-characterized role played by RBM20 in the regulation of Titin splicing [144]. Titin encodes an elastic protein in cardiac muscle cells, which contributes to ventricular wall stiffness and Titin mis-splicing, which as a consequence of reduced RBM20 expression or function, is associated with heart failure [144]. Thus, it is possible that dysregulation of the PI3K pathway in cardiomyocytes could more generally alter the physiology of the organ through changes in RBM20-mediated splicing of muscle-specific genes.

Another interesting example of how the PI3K/AKT pathway impacts RNA splicing is illustrated by the work of Feng and colleagues. PRMT6-mediated methylation of arginine 159 in the amino-terminal phosphatase domain of the phosphatase and tensin homolog (PTEN) is critical for its phosphatase activity and for its function as a negative regulator of the PI3K/AKT cascade [145]. Transcriptome analyses of PTEN-null H4 glioma cells, in which either the wild-type PTEN or a methylation-defective (R159) mutant were overexpressed, uncovered massive AS dysregulation as a consequence of the constitutive activation of the PI3K/AKT signaling pathway in the PTEN mutant cells [145]. Since the PI3K inhibitor LY294002 mimicked regulation by wild-type PTEN, it is conceivable that PTEN-dependent regulation of AS relies on suppression of the PI3K/AKT cascade [145]. Since the PTEN R159K mutation has been found in numerous somatic human cancers, including glioma, melanoma and thyroid cancer [145], the consequent global deregulation of splicing might contribute to the oncogenic program triggered by the constitutive activation of the PI3K/AKT pathway. These examples highlight the key role played by the PI3K/AKT pathway on direct and indirect regulation of RNA processing and suggest that these post-transcriptional mechanisms contribute to the effects elicited by this pathway on multiple physiological and pathological processes.

### 5.2. The RAS/MAPK Pathway and RNA Processing Regulation

The small G protein RAS controls a key mitogenic pathway that relies on the evolutionary conserved RAF/MEK/ERK kinase cascade. The RAS pathway leads to activation of the mitogen activated protein kinases (MAPKs) ERK1 and 2 and couples signals from several cell surface receptors with intracellular events that ultimately regulate gene expression [146]. Some components of this pathway are frequently mutated or aberrantly expressed in human cancer, leading to uncontrolled proliferation and tumorigenesis [146]. The oncogenic role of the RAS/MAPK pathway is due, at least in part, to RNA splicing/processing regulation in favor of pro-tumoral isoforms. One of the first experimental evidences in this sense was the observation that, upon T-cell activation, the RAS cascade positively regulated the inclusion of variable exons in the mature mRNA of the CD44 gene [74,147,148]. The CD44 gene encodes for 10 variable exons localized in the middle of the transcription unit, which are flanked by constitutive exons on both sides. A positive feedback loop was shown to couple RAS activation and alternative splicing of CD44 isoforms containing the variable exons. Growth factor-dependent activation of the RAS pathway promotes the expression of exon v6-containing CD44 isoforms which, in turn, sustained late RAS signaling [149]. Such a positive feedback loop could play a functional role in the transition from normal to transformed phenotypes by also maintaining constitutive activation of the RAS pathway in tumors that lack oncogenic RAS mutations. Mechanistically, AS regulation of CD44 by the RAS signaling cascade mostly relies on the increased phosphorylation of some splicing factors. ERK1/2-mediated phosphorylation of Sam68 promoted the inclusion of variable exons in the mature CD44 transcript [74,147,148]. Moreover, Sam68 interacts and cooperates with the spliceosome component SRm160 and the chromatin remodeling protein BRM to promote splicing of CD44 variable exons downstream of RAS pathway activation [74,147,148].

A similar scenario was described in colorectal cancer cells undergoing epithelial-to-mesenchymal transition. By switching them from high to low-density cultures, these cells acquire a mesenchymal phenotype and activate ERK1/2, which in turn phosphorylate and activate Sam68. This RBP binds to and promotes the retention of an intronic sequence in the 3′UTR of the SRSF1 mRNA, preventing its degradation through the nonsense-mediated decay pathway [150]. Thus, ERK1/2 activation results in increased SRSF1 expression and SRSF1-mediated splicing of pro-mesenchymal isoforms, such as the ∆RON variant of the proto-oncogene RON [150]. Furthermore, the RAS pathway also stimulates Sam68-mediated splicing of the oncogenic cyclin D1b variant in prostate cancer cells [87], further pointing to Sam68 as a key effector of the splicing program activated by this pathway.

In addition to Sam68, activation of ERKs was also reported to regulate the activity of other splicing factors. Upon oxidative stress, these kinases phosphorylate SPF45 on threonine 71 and serine 222, leading to repression of exon 6 inclusion in the FAS mRNA and to production of a dominant negative isoform of this death receptor [151]. Furthermore, ERK-dependent phosphorylation of the carboxyl-terminal domain of DAZ-associated protein 1 (DAZAP1) affects its splicing activity by regulating its subcellular localization [152]. Upon activation of the RAS pathway, DAZAP1 translocates into the nucleus and regulates an AS program by directly binding to cis-acting splicing regulatory elements and by competing with the splicing repressor hnRNP A1 [152]. The endogenous targets of DAZAP1 were found to be significantly enriched in cell cycle-related genes, suggesting that DAZAP1-mediated splicing is part of the mitogenic program regulated by the RAS/MAPK pathway [146,152].

The RAS pathway can also modulate splicing decisions by affecting the activity of co-regulators of RBP functions. The scaffold/matrix-associated region-binding protein 1 (SMAR1) binds to and represses recognition of variable exons v3 and v5 in the CD44 pre-mRNA [153]. SMAR1 exerts this role as part of an RNA-dependent trimeric complex that also comprises Sam68 and HDAC6, maintaining Sam68 in a deacetylated state. Upon activation of the RAS/MAPK pathway, ERK-dependent phosphorylation of SMAR1 at threonine 345 and 360 leads to its translocation in the cytoplasm (Figure 2B). Shuttling of SMAR1 protein from nucleus to cytoplasm disrupts the trimeric complex on the CD44 pre-mRNA and favors Sam68 acetylation [153]. Since this PTM enhances the affinity of Sam68 for RNA [74,76], RAS activation ultimately leads to increased expression of CD44 variants comprising the variable exons, which confer an invasive and metastatic phenotype to breast tumor cells (Figure 2B) [153]. Interestingly, since both SMAR1 [153] and Sam68 [148,154] interact with the snRNPs, the trimeric SMAR1–Sam68–HDAC6 complex might also act as a roadblock to sequester snRNPs and to prevent their functional interaction within the spliceosome.

### 5.3. RNA Processing Regulation in Response to Heat Shock

In response to various environmental stresses, such as hyperthermia, ischemia and anoxia, cells induce the synthesis of a small group of proteins named “heat shock” proteins (HSPs), which represents a natural defense response to cope with hostile conditions. HSPs are expressed in all organisms and are highly conserved and ubiquitous [155]. They play a pivotal role as molecular chaperones to acquire and maintain the innate structures and functions of their target proteins but are also involved in several processes like protein secretion, transport, translocation, degradation and gene regulation [156]. However, in spite of their generally protective functions, dysregulation of HSP expression can contribute to the development of several diseases, including cancer [156]. 

In the absence of stress, heat shock transcription factor 1 (HSF1) is sequestered and repressed by interactions with HSP90 and HSP70 in the cytoplasm. Upon stress, HSPs dissociate from HSF1 and allow it to translocate into the nucleus and activate the transcription of HSP genes by binding to specific heat shock elements (HSE) in their promoter regions. Conversely, the transcription, splicing and translation of non-HSP genes are repressed. Indeed, heat shock was found to cause widespread retention of introns in thousands of transcripts, leading to their accumulation in the nucleus and preventing their translation. However, genes encoding for proteins in oxidation reduction and protein-folding functions continued to be efficiently spliced [157]. Thus, global repression of splicing is a rapid response of mammalian cells to heat stress, which probably aims at saving energies under difficult conditions. Splicing repression upon heat stress is mainly caused by a rapid dephosphorylation of SRSF10 (also known as SRp38), which sequesters the U1 snRNP [158]. Moreover, the tri-snRNPs (U4/U5/U6) are dissociated and several splicing factors are sequestered in nuclear stress bodies [159,160,161,162]. However, the repression of splicing is not complete, as processing of HSP90 transcripts is functional while that of HSP27 is only partially inhibited [163]. HSP27 plays an active role in the regulation of splicing during heat shock. This small protein is phosphorylated at serine 15, 78 and 82 by the p38 MAPK/MK2 module and its phosphorylation increases the extent of splicing during recovery from heat shock by promoting the re-phosphorylation and inactivation of SRSF10 [164]. Since pharmacologic inhibition of HSP90 blocked the re-phosphorylation of SRSF10 and the recovery of splicing after heat shock, HSP90 is also likely necessary for this process [164]. Subsequent studies confirmed the role of HSP27 in splicing regulation and showed that a large number of genes are susceptible to HSP27 depletion [165].

More recently, a role in splicing regulation was also described for DNAJC17, a member of the HSP40 family [166]. DNAJC17 is an essential protein that was identified as a susceptibility factor for congenital hypothyroidism and myeloproliferative disorders. Nevertheless, its specific biological functions are poorly known. High-throughput transcriptomic and proteomic approaches have recently highlighted a link between DNAJC17 and a network of proteins involved in splicing. This small HSP interacts with several splicing factors and localizes within nuclear speckles that are enriched in spliceosomal components. Moreover, DNAJC17 depletion induced widespread changes in AS of endogenous genes in HeLa cells. These findings uncovered a novel role for DNAJC17 in splicing-related processes and suggest that splicing impairment may contribute to its essential function in early development [166].

Interestingly, a splicing response to heat shock is also present in plants. High temperatures disturb cellular homeostasis and growth in plants, which have developed specific responses to resist to heat stress. Among other responses, heat shock also impacts AS regulation, which contributes to thermotolerance [167]. An example is provided by *Arabidopsis thaliana*. During the co-evolution of eukaryotic host cells and α-proteobacteria, plants have developed a specific regulatory mechanism named mitochondrial intron splicing. The removal of selected introns is favored by mitochondrial-targeted proteins that are encoded in the nucleus. One of these proteins in *Arabidopsis thaliana* is WTF9 (what’s this factor 9), which is involved in the splicing of two mitochondrial-encoded genes named rpl2 and ccmFC [168]. Co-immunoprecipitation and pull-down assays demonstrated that HSP60 proteins interact with WTF9 and with a 48 nucleotide-long region in the ccmFC intron. HSP60 inactivation results in reduction of the splicing efficiency of the rpl2 and ccmFC pre-mRNAs and a small size phenotype of the plant. Since the same phenotype was also observed in the absence of WTF9, it was suggested that, through their RNA-binding ability, HSP60 proteins play a role in splicing of rpl2 and ccmFC introns in the mitochondria [168]. Furthermore, the heat shock transcription factor A2 (HsfA2), a key regulator of the response to heat stress in this plant, is regulated by AS during heat shock [169]. The heat stress-induced splice variant (HsfA2-III) is generated through a cryptic 5’ splice site in the intron and encodes for a truncated protein that is involved in auto-regulation of HsfA2 transcription. Since splicing of other Hsf genes was also regulated by heat shock, this study revealed a key role for RNA processing regulation in the orchestration of the response of *Arabidopsis thaliana* to heat stress [169].

### 5.4. The DNA Damage Response Pathway and RNA Processing Regulation

DNA damage is another stress that globally affects RNA processing. Eukaryotic cells are endowed with a DNA damage response (DDR) signaling pathway that helps maintain genome stability under DNA-damaging stress [170,171]. The DDR pathway senses the DNA damage, halts cell cycle progression and promotes the repair of the lesion. To properly execute these actions in time and space, the DDR also needs to coordinate the regulation of gene expression at the transcriptional and post-transcriptional level [172,173]. If the damage is too severe and/or incorrectly solved, induction of programmed cell death prevents the propagation of the defect to daughter cells upon division, which could lead to accumulation of mutations and to the onset of pathological conditions, such as neoplastic transformation [170,171].

The main players of the DDR pathway are represented by three protein kinases-ataxia telangiectasia and Rad3-related protein (ATR), ataxia-telangiectasia mutated (ATM) and DNA-dependent protein kinase (DNA-PK), and by their downstream effectors, such as checkpoint kinase 1 (CHK1) and CHK2 [170,171]. These DDR protein kinases phosphorylate hundreds of substrates [173], including RNAPII, splicing factors and other RBPs involved in RNA processing regulation [174]. Thus, it is not surprising that activation of the DDR pathway has an impact on both transcriptional and post-transcriptional steps involved in the regulation of gene expression. DNA damage, induced by exposure to ultraviolet irradiation (UV), was shown to affect the phosphorylation status of the carboxy terminal domain (CTD) of RNAPII [175]—this PTM elicited strong effects on the transcriptional elongation rate as well as on AS and APA regulation [176]. Hyperphosphorylation of the CTD in response to UV reduced the transcriptional elongation rate; this event was associated with AS of multiple genes that are sensitive to the dynamics of RNAPII [175]. Subsequent studies showed that single-stranded DNA (ssDNA) that is exposed during nucleotide excision repair (NER) of pyrimidine dimers—the most conspicuous UV-induced DNA lesion—leads to activation of ATR that, in turn, promotes CTD phosphorylation [177]. The regulation of this PTM is probably indirect, as the CTD sequence lacks recognizable ATR consensus motifs. Thus, ATR likely acts by signaling the DNA lesions and linking their repair to regulation of RNAPII function [177]. Notably, many genes involved in apoptosis, cell cycle progression and the DDR pathway were regulated by splicing in response to UV [175,177]. Furthermore, the reduced elongation rate of RNAPII caused by UV-induced DNA damage was shown to trigger the formation of short, non-coding transcripts from protein-encoding pre-mRNAs as result of widespread selection of proximal alternative last exons [178]. This process may represent an adaptive response to genotoxic stress, as indicated by the shorter non-coding isoform of the activating signal cointegrator 1 complex subunit 3 (ASCC3) gene that is directly involved in the recovery of transcription after UV irradiation (Figure 3A) [178].

Widespread modulation of AS is also induced by transcription-inhibiting DNA damage. Stalling of the RNAPII, as a consequence of a DNA lesion, was shown to promote the displacement of late-stage spliceosomal subunits (U2/U5/U6 snRNPs) from the chromatin and the formation of RNA:DNA hybrid structures known as R-loops. Next, the R-loops activated ATM that further blocked the recruitment of the spliceosome and led to global changes in splicing regulation [179]. Interestingly, this study proposed that splicing and ATM are subjected to a reciprocal regulation, whereby changes in the organization of the splicing machinery activates ATM signaling and, in turn, ATM regulates AS of select genes [179]. Thus, splicing regulation represents one of the main cellular responses to DNA damage and different genotoxic stresses can modulate this process by activating separate, yet communicating, DDR pathways.

Genotoxic stress can also affect RNA processing regulation by altering the function or the expression levels of RBPs [180]. ATM-dependent phosphorylation of HuR upon oxidative stress increased binding of this RBP to TRA2β pre-mRNA and promoted exon 2 inclusion. The resulting splice variant comprises multiple premature stop codons and encodes for a shorter isoform, thus reducing the expression levels of the canonical TRA2β protein [181]. Moreover, DNA-PK was also reported to phosphorylate several hnRNPs and the DNA/RNA helicase DHX9 [182]. This PTM may play a functional role in the recently described modulation of AS by DNA-PK in response to formation of DNA double strand breaks [183]. Following its dissociation from the DNA lesions caused by mitoxantrone, DNA-PK accumulates in nuclear speckles enriched in splicing factors. Intriguingly, inactivation of DNA-PK, by either chemical inhibition or RNA interference, affected splicing of several RBPs, including SRSF1, SRSF2, hnRNP DL, hnRNP H1 and PRPF38B [183]. Although the specific mechanism was not investigated, this experimental evidence strongly suggests that DNA-PK is also directly involved in AS regulation during genotoxic stress. 

In other cellular contexts, DNA damage affected the expression level of specific RBPs. As an example, the expression of SRSF1 is induced by UV irradiation and gemcitabine treatment in cancer cells, where it regulates splicing events that promote survival to these genotoxic stresses [184,185,186]. Moreover, the subnuclear distribution of several RBPs, such as Sam68, EWS and SRSF1 is regulated upon DNA damage, with consequent impacts on AS of their target genes [187,188,189]. In particular, it was shown that UV-induced translocation of EWS changes the splicing pattern of genes involved in DNA repair and genotoxic stress signaling, including ABL1, CHEK2 and MAP4K2 [188]. DNA damage was also shown to affect the functional interaction between RBPs. Treatments with both camptothecin and cisplatin disrupt the interaction of EWS with the spliceosome-associated YB-1 protein, causing the skipping of several exons of the MDM2 gene. MDM2 is the main E3 ubiquitin–protein ligase that controls p53 degradation and this splicing event reduces MDM2 expression and contributes to the accumulation of p53 during genotoxic stress (Figure 3B) [190]. A similar mechanism controls the function of the SR protein SRSF10, which forms a trimeric complex with hnRNP K and hnRNP F/H to regulate the splicing of the apoptotic BCL-X gene (BCL2L1) [191]. Upon oxaliplatin-induced DNA damage, dephosphorylation of SRSF10 and hnRNP K impairs their interaction with both BCL-X pre-mRNA and hnRNP F/H, allowing the latter RBP to promote the pro-apoptotic BCL-Xs splice variant [191]. This mechanism is also consistent with the previously reported effect of protein phosphatases on oxaliplatin-induced BCL-X splicing [192].

As described for splicing, pre-mRNA 3′-end processing is also generally inhibited during the DDR and this block contributes to the general inhibition of protein expression in response to genotoxic stresses [193]. Indeed, unprocessed transcripts lacking a polyA tail at the 3′-end are generally degraded in the nucleus or are unable to be transported to the cytoplasm for translation. However, compensatory mechanisms allow proper processing of specific transcripts that are functionally requested by the cell during the recovery from DNA damage, like that encoding the p53 tumor suppressor protein. In this context, hnRNP F/H and the RNA helicase DHX36 were shown to bind to an RNA G-quadruplex structure located in the vicinity of a polyadenylation site in the p53 transcript [194,195]. These RBP–RNA interactions are critical for the accumulation of p53 protein during stress and likely contribute to p53-mediated apoptosis. Furthermore, mounting evidence points to additional direct links between the DDR pathway and RBPs involved in 3′-end processing and APA regulation. For instance, the nuclear poly(A)-binding protein 1 (PABPN1) is a target of ATM and plays a direct role in the DDR [196]. PABPN1 is recruited to sites of double strand breaks and is required for their optimal repair. Notably, this study also identified several other RBPs involved in RNA processing that are recruited to sites of DNA lesions, which may be required for the processing of the newly discovered small non-coding RNAs that are an integral part of the DDR pathway [197]. Likewise, CSTF50, a core component of the CSTF complex involved in APA regulation, is a cofactor of the BRCA1/BARD1 complex that facilitates chromatin remodeling during the DDR. CSTF50 may help bridging the RNAPII to the BRCA1/BARD1 complex and to promote its ubiquitination [198], an event that contributes to transcription-coupled repair during the DDR [199].

Collectively, these studies confirm the existence of an extensive interface between the RNA metabolism and DDR pathways [180] and suggest that widespread modulation of AS and APA is an integral part of the cellular response to genotoxic stresses [193].

### 5.5. The Circadian Clock Pathway and RNA Processing Regulation

Circadian rhythmic changes in light and temperature force living organisms to adapt their functions to these daily changes. The so-called circadian clock is a molecular mechanism that regulates behavior, metabolism and physiology cycles on the basis of light availability [200,201,202]. In mammals, sleep/activity alternation, body temperature fluctuation, hormone levels and metabolism are all governed by 24 h cycles orchestrated by circadian rhythms, even in the absence of other external cues. Disruption of such circadian homeostasis can, in the long term, contribute to disease development [200,201,202]. The central timekeeper is the suprachiasmatic nucleus of the hypothalamus (SCN), which receives the light stimulus and activates the response. The SCN also acts as a pacemaker to synchronize other peripheral clocks [203,204]. Neural networks, hormones (glucocorticoids) and behavioral pathways (sleep/wake and food intake) all contribute to the transmission of the timing signals from the SCN to peripheral tissue [200,201,202,203,204]. In particular, synaptic inputs from intrinsically photosensitive retinal ganglion cells (ipRGCs) allow neurons in the core region to maintain synchrony within the SCN [205]. Interestingly, the liver also functions as an independent circadian oscillator, with timing of food intake as the primary starting signal, even though the SCN is also important for liver rhythmic gene expression [206]. 

Despite differences in the circadian clock machinery that exist among organisms, the circadian cycle depends on the transcription of a core set of clock genes that are conserved across species [207]. The circadian clock relies on a transcriptional–translational feedback loop whose period length is dictated by PTMs [208,209]. Circadian transcription is regulated by the heterodimer formed by CLOCK (circadian locomotor output cycles kaput) and BMAL1 (brain and muscle ARNT-like 1). These transcription factors bind to E-box enhancers and induce the expression of specific targets, like the period (PER1 and PER2) and cryptochrome (CRY1 and CRY2) genes. The newly synthesized PER and CRY proteins translocate into the nucleus and form a heterodimer that inhibits the transcriptional activity of CLOCK:BMAL1 by direct interaction (Figure 4) [200,205]. In the meanwhile, the CLOCK/BMAL1 dimer also induces the expression of the nuclear receptor REV-ERBα, which acts in a negative feedback loop by binding to and repressing retinoic acid receptor-related orphan receptor binding element (RRE) in the BMAL1 promoter (Figure 4A). The activity and stability of the PER/CRY complexes are then controlled by PTMs that promote their inactivation and degradation and untether the CLOCK:BMAL1 complex for the beginning of a new cycle [210,211,212]. Importantly, PTMs, and in particular phosphorylation, of circadian regulators play a central role in the clock and alternation of activating/inactivating modifications of several players involved in the clock timing, occurring in a rhythmic fashion during the 24 h cycle [210,211,212].

In addition to modulating the expression of target genes, mounting evidence indicates that the circadian rhythm also globally affects AS and APA regulation. Splicing regulation of many genes was shown to follow a circadian rhythm in the mouse liver [213]. Moreover, expression of a subset of RBPs also cycled in the mouse liver, including AS regulators such as SRSF3, SRSF5, TRA2β, SF3B1, hnRNPs and the RNA helicases DDX46 and DHX9, which likely contributed to regulation of cycling exons [213]. An additional functional connection between AS regulation and the circadian clock was uncovered when it was documented that exons 6 and 7 of the mouse U2af26 gene undergo a splicing switch that follows circadian and light rhythms [214]. This splicing event changes the reading frame of the transcript and introduces a domain homologous with part of the TIMELESS protein of *Drosophila* changes in the carboxyl-terminal region of the U2AF26 protein. The resulting isoform interacts with PER1 and modifies its stability, thus affecting clock functions (Figure 4A). U2af26 knockout mice have arrhythmic PER1 protein levels and display defects in the expression of circadian genes. Interestingly, these mice also exhibit an increased adaptation capacity to experimental jet lag, suggesting a functional role of U2AF26 splicing in avoiding abnormal changes in the circadian clock under light/dark conditions [214]. Shortly after, it was also proposed that body temperature cycles are necessary and sufficient for driving rhythmic SR protein phosphorylation, which controls AS during the circadian cycle [215]. Temperature changes of as little as 1 °C were sufficient to cause a splicing switch in several genes, including U2af26 and Tbp, encoding the TATA-box binding protein. Importantly, rhythmic Tbp splicing affects global gene expression during the cycle, indicating that even a moderate alteration in temperature can control widespread changes in the transcriptome through a splicing switch [215]. 

Changes in the polyadenylation pattern during the circadian cycle may also contribute to clock control in mammalian cells. A long poly(A) tail generally stabilizes the mRNA, whereas its shortening promotes transcript degradation. Notably, the poly(A) tail length of some circadian genes was shown to undergo cycling regulation, due to rhythmic expression of the deadenylase nocturnin (Noct gene) in various murine tissues, with peak levels at the time of light offset. Since NOCT expression is induced by serum shock, extracellular signals may determine the timing of gene silencing through activation of this deadenylase [216,217]. More than 2% of the transcripts expressed in the mouse liver undergo rhythmic modification of their poly(A) tail length [218]. These poly(A)-rhythmic mRNAs display peak tail lengths during all phases of the daily cycle, even though a significantly higher number of transcripts are characterized by long/short peak ratios during the night. Importantly, the rhythmicity in poly(A) tail length correlated with the levels of protein expression a few hours later [218], suggesting that circadian changes in transcript polyadenylation can determine rhythmic protein expression in the cell.

The site of mRNA polyadenylation can also be under circadian control and determine changes in protein functions during the daily cycle [219], as APA isoforms with different 3′UTR lengths can be differentially recognized by microRNAs and RBPs that modulate mRNA stability, translation and subcellular localization [14,15,21]. Temperature-dependent upregulation of the cold-inducible RNA-binding protein (CIRBP) and RBM3 influence the selection of the pA site, resulting in the expression of stable transcripts characterized by a longer 3′UTR. Such APA switches are dictated by the circadian rhythm and regulate many transcripts in the liver [220]. 

The circadian clock has a central role in modulating plant responses to environmental cues. For this reason, selection of circadian clock variants has been implicated in adaptation and domestication of many species of great importance in agriculture [221,222]. Transcriptome analyses in seedlings grown under the “long day condition” (16/8 h light/dark) or under constant light for two days highlighted circadian genes that are expressed rhythmically even under constant light in *Arabidopsis thaliana*. These studies uncovered extensive light-sensitive AS and APA programs in this plant. Of note, such rhythmic patterns were correlated with gene-level oscillations of factors involved in AS and APA [223,224]. More recently, it was shown that the TOR kinase, homolog of mammalian mTOR, modulates a specific splicing program in response to light. Interestingly, however, these splicing changes were observed in the root, which is not directly exposed to light. However, light-induced retrograde signals regulate the expression of splicing-related factors in roots through activation of TOR. Indeed, chloroplasts perceive light in the leaves, activate photosynthesis and produce sugars that are transferred to the root. In the root, sugars are metabolized to pyruvate, which is transferred to mitochondria and used for oxidative phosphorylation, leading to activation of TOR and regulation of the light-induced splicing program [225]. These findings suggest that light can also exert long-distance effects on splicing through synthesis and mobilization of sugars and activation of the TOR pathway throughout the whole plant. Thus, regulation of RNA processing mechanisms by light and/or the circadian clock also plays a crucial role in plant physiology.

## 6. RNA-Based Therapies as Tools to Correct Disease-Related Signal Transduction Pathways

As described above, AS and APA increase the diversity and plasticity of the coding potential of eukaryotic genomes. However, their flexibility is prone to errors and alteration of these mechanisms often contributes to disease onset and/or progression [20,21]. Thus, understanding the mechanisms that regulate these RNA processing events and how they are altered in disease conditions may pave the ground for novel targeted therapies that correct specific molecular defects. In this regard, several studied have revealed how splicing is pathologically altered [226]. Among other mechanisms, mutations affecting splicing regulatory sequences of critical cancer-associated genes and mutation or gene expression alterations affecting core and accessory components of the spliceosome complex frequently occur in human cancers [227]. Since AS and APA regulation involves base pair recognition by regulatory RNAs and/or proteins, it offers the opportunity to design therapeutic tools with elevated target selectivity. A remarkable example of the power of RNA-based therapies is provided by Nusinersen, a drug recently approved for treatment of spinal muscular atrophy (SMA) patients [228]. In humans, the essential SMN protein is encoded by the two highly homologous SMN1 and SMN2 genes. The disease is caused by inactivating mutations in the SMN1 gene. Although patients maintain an intact SMN2 gene, a single silent transition (C6->T) in exon 7 leads to its being skipped in the majority of transcripts and production of a shorter, unstable SMN2 protein. A splicing silencer element in intron 7 of the SMN2 gene was identified as a crucial determinant of increased exon 7 skipping, which acts by recruiting splicing-repressing RBPs. The sequence of this regulatory element was exploited to develop a splice-switching antisense oligonucleotide (ASO) that counteracts the effect of this repressive element and rescues exon 7 splicing, SMN protein expression and SMA phenotypes in animal models and patients [228]. 

ASO technology could also be applied to human diseases in which splicing affects the function of key signal transduction proteins. In this regard, a very recent work showed that the U2AF65-related protein RBM39, induced by treatment with an antineoplastic sulfonamide compound named Indisulam, selectively repressed expression of the splice variant A (KRAS4A) of the frequently mutated KRAS oncogene, which is enriched in cancer cells with stem-like properties. Importantly, chemical inhibition of KRAS4A splicing eliminated cancer stem cells and improved disease outcomes in an animal model [229]. Since expression of the KRAS4A^G12V^ mutant isoform was recently shown to induce metastatic lung adenocarcinomas [230], development of ASOs that specifically target this splice variant could represent an excellent strategy for cancers that rely on mutations in the KRAS oncogene and activation of its downstream pathway, such as the highly aggressive pancreatic ductal adenocarcinoma (PDAC) or lung cancer. Notably, widespread alterations in splicing are caused by a frequently occurring mutation in the TP53 gene in PDAC, and specific aberrant splice variants were shown to further activate the oncogenic KRAS signaling pathway [231]. Suppression of this aberrant splicing program by treatment with a spliceosome inhibitor was shown to slow disease progression in PDAC mouse models [231], indicating a clear synergism between splicing and signaling alterations in this cancer. Thus, ASOs, that specifically target the splice variants identified as responsible for the oncogenic effects, would also have the advantage of avoiding widespread changes in splicing regulation which are induced by chemical inhibitors of splicing regulators, like Indisulam or Pladienolide B derivatives [229,231]. 

On the other hand, targeting PTMs or signaling pathways impinging on RNA processing mechanisms that are dysregulated in human diseases could also represent a valuable therapeutic strategy. For instance, inhibition of protein phosphatases was investigated as a therapeutic option for SMA. Indeed, the SR-like splicing factor TRA2β, which promotes splicing of SMN2 exon 7 [232], interacts with PP1 through its RRM. Dephosphorylation of TRA2β by PP1 represses its activity. Conversely, chemical inhibition of PP1 promotes the inclusion of TRA2β-target exons, including SMN2 exon 7 [233], suggesting the possible therapeutic potential of this strategy for SMA. However, PP1 inhibitors affect a wide spectrum of targets and may be detrimental for the cell. Thus, strategies that allow selective delivery of such inhibitors to the specific substrate that is being targeted should be developed. 

Splicing-modulating drugs could also be used to enhance the efficacy of treatments that target signal transduction pathways. Dysregulation of the mTOR complex 1 (mTORC1)-dependent pathway frequently occurs in pancreatic neuroendocrine neoplasms (PanNEN); the mTORC1 inhibitor Everolimus is used as a therapeutic option for the treatment of these tumors. Unfortunately, after initial remission, most patients experience disease progression due to acquired resistance to the treatment. A recent study has shown that upregulation of MYC contributes to such secondary resistance, and that combined treatment with Dinaciclib, a pan-CDK inhibitor that also targets kinases involved in RNA processing regulation such as CDK12 [234], re-established sensitivity to Everolimus [235]. Another example of combined treatment was proposed for Ewing sarcoma [142], where most patients acquire resistance or do not respond to the currently used chemotherapeutic treatments. Inhibition of the PI3K/AKT/mTOR axis, which is often deregulated in this tumor, is considered a potentially valuable therapeutic strategy. However, inhibitors of this pathway cause widespread modulation of AS events, which is mediated by hnRNPM upregulation and promotes resistance to the treatment. On this basis, it was suggested that inactivation of hnRNPM or reversion of disease-relevant splicing events regulated by this RBP could be exploited to increase the efficacy of inhibitors of the PI3K/AKT/mTOR pathway in this tumor type [142].

## 7. Concluding Remarks and Future Perspectives

Guaranteeing cellular homeostasis is fundamental to ensuring the survival of eukaryotic cells. To this aim, specific stimulus-dependent signaling pathways are activated in the proper time windows to respond to environmental cues. These pathways allow the coordination of the activity of many proteins and RNA molecules that are already present in the cell, as well the induction of the expression of other molecules that are required only under specific conditions. RBPs represent one of the most abundant classes of substrates of such signaling cascades, and PTMs elicited by enzymes in these pathways modulate their expression, localization and activity—thus influencing the multiple steps involved in the RNA maturation process. However, our current knowledge about cell signaling-dependent regulation of RNA processing is likely the tip of the iceberg. The number of RBPs and their involvement in many different cellular processes is expanding [236], adding new layers of complexity to the already intricate interplay between signal transduction pathways and RNA metabolism. Furthermore, genome-wide transcriptome analyses have now revealed the existence of an enormous amount of non-coding RNA molecules that are actively transcribed by eukaryotic genomes, whose contribution to the modulation of signal transduction pathways is only beginning to emerge. Likewise, non-canonical transcripts, such as circular and chimeric RNAs, are also extensively produced by typical protein coding genes through back-splicing or trans-splicing mechanisms [237,238]. While these non-canonical RNA molecules are emerging as new regulators of gene expression [239], if and how they are regulated by signaling pathways in response to specific stimuli is not well described yet.

Disruption of the crosstalk between signaling pathways and RNA processing is frequently associated with pathological conditions, such as neurodegenerative diseases and cancer. Thus, deepening our knowledge about this interplay is of primary importance to prevent the onset and progression of human diseases. Furthermore, understanding the defective mechanisms that promote disease is key to paving the ground for the development of targeted treatments. One clear example in this sense is provided by TDP-43, a multifunctional RBP subjected to various types of PTMs whose functional dysregulation is strongly associated with frontotemporal dementia and ALS. These neurodegenerative diseases share the common feature of cytoplasmic aggregates marked by accumulation of TDP-43, which then leads to neuronal death. Since TDP-43 present in aggregates often displays increased PTMs, understanding whether and how these modifications contribute to its toxicity, and elucidating the signaling pathways that contribute to these PTMs, may offer new therapeutic perspectives to these currently non-curable diseases [240].

Some efforts to develop efficient therapeutic tools targeting the splicing activity of specific RBPs or the activity of disease-related splice variants have been made recently [82,227]. However, in the future it would be desirable to combine these approaches with drugs that target signaling molecules and assess the potential synergism between these treatments. Moreover, since aberrant processing of transcripts encoding signaling proteins can generate constitutively active isoforms and soluble truncated isoforms acting in a dominant-negative manner [240], RNA-based therapies correcting these defects may improve the efficacy of standard therapies already in use. In this scenario, although the existence of feedback loops between signal transduction pathways and RNA processing makes understanding of the regulatory mechanisms involved more complex, these links may also be exploited to develop novel therapeutic options.

## Figures and Tables

**Figure 1 biomolecules-11-01475-f001:**
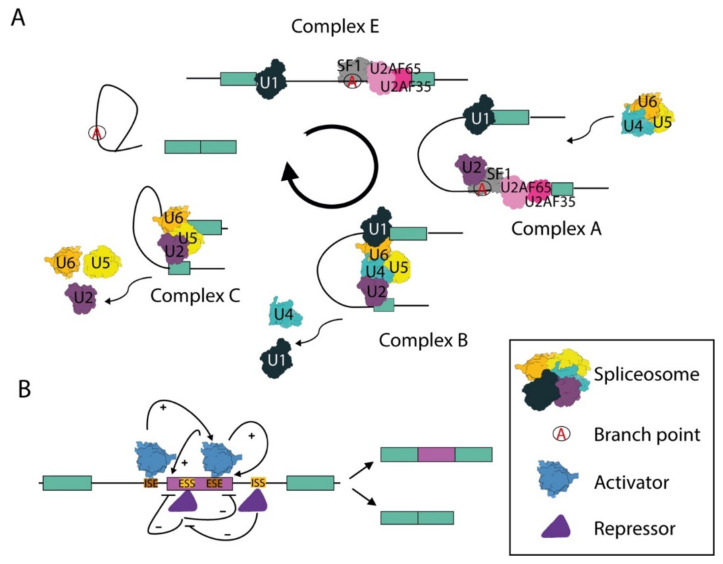
The splicing mechanism. (**A**) Schematic representation of the splicing reaction. In the first step, the U1 snRNP binds to the 5′ splice site whereas SF1 and the U2AF complex bind the branch-point and the 3′ splice site, respectively, resulting in the formation of the E complex. The subsequent binding of U2 snRNP at the branch-point, which replaces SF1, leads to the formation of the A complex (or pre-spliceosome). After recruitment of the tri snRNP U4/U6 and U5, the spliceosome assumes its B complex conformation, which is then remodeled through numerous protein–protein and protein–RNA interactions to acquire the catalytically active conformation (complex C) that promotes intron excision and exon joining. (**B**) Alternative splicing of variable exons is regulated by the presence of intronic and exonic splicing enhancers (ISE and ESE), which are bound by splicing activators (blue) that promote recognition of the splice sites by the spliceosome and the inclusion of the alternative exon (pink). Conversely, splicing repressors (purple triangles) bind to intronic and exonic splicing silencers (ISS and ESS) and repress inclusion of the alternative exon by hiding the splice sites and competing with the spliceosome.

**Figure 2 biomolecules-11-01475-f002:**
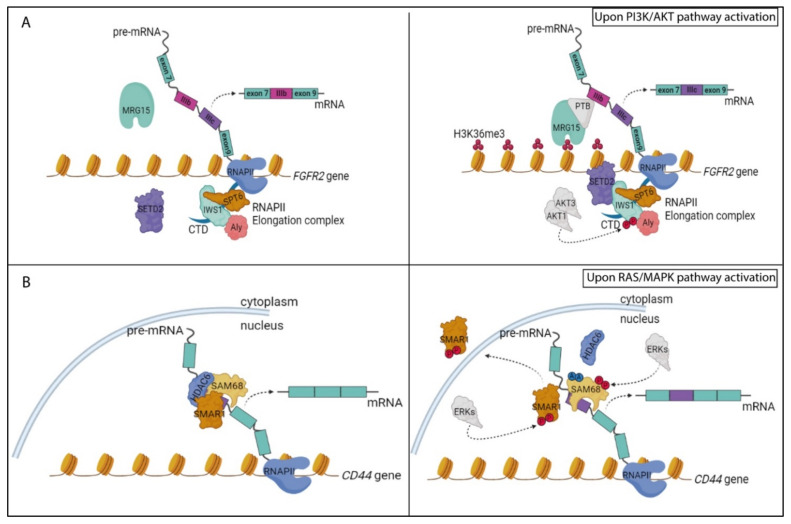
Examples of signaling-dependent regulation of alternative splicing. (**A**) The two alternatively spliced isoforms of FGFR-2 comprise one of the two mutually exclusive exons IIIb and IIIc. In healthy cells, FGFR-2 mature mRNA mainly contains the variable exon IIIb. However, AKT1/3-mediated phosphorylation of IWS1 in lung carcinoma cells favors the recruitment of the histone H3K36 methyl transferase SETD2 to the carboxyl-terminal domain (CTD) of RNA polymerase II (RNAPII). SPT6 and Aly are part of the RNAPII elongation complex together with IWS1. SETD2-dependent trimethylation of H3K36 (H3K36me3) of exon IIIb promotes the recruitment of the chromatin-binding protein MRG15 and of the splicing factor PTBP1, which repress the inclusion of exon IIIb in the mature transcript and lead to splicing of exon IIIc. (**B**) In healthy cells, SMAR1 is located in the nucleus and is part of an RNA-dependent trimeric complex together with Sam68 and HDAC6. The complex is bound to alternative exons (in purple) of CD44 pre-mRNA, thus repressing their inclusion into the mature CD44 mRNA. In breast tumor cells, activation of the RAS/MAPK pathway results in ERK1/2-dependent phosphorylation of SMAR1 and Sam68, leading to the translocation of the nuclear protein SMAR1 into the cytoplasm, the release of HDAC6 into the nucleoplasm, and the acetylation of Sam68. In turn, acetylation and phosphorylation of Sam68 induces its splicing activity and promotes the inclusion of the alternative exons into the mature CD44 mRNA.

**Figure 3 biomolecules-11-01475-f003:**
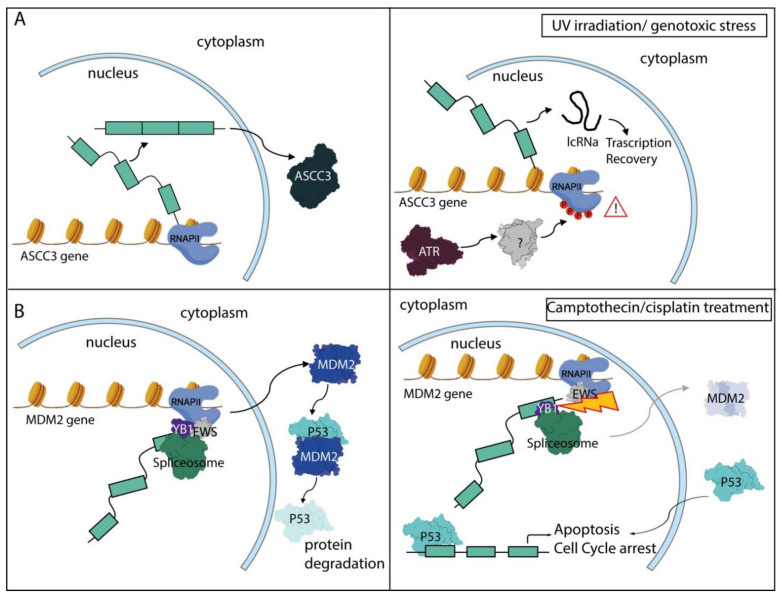
Regulation of splicing by the DNA damage pathway. (**A**) DNA damage caused by UV irradiation activates the ATR kinase, which promotes the hyperphosphorylation of the CTD domain of RNAPII, thus reducing its elongation rate. A slow elongation rate favors the premature termination of the pre-mRNA and production of a short and non-coding transcript, as in the case of ASCC3 (activating signal cointegrator complex subunit 3). The short, non-coding ASCC3 then induces the recovery of transcription after resolution of the DNA lesion. (**B**) Treatment with camptothecin or cisplatin disrupt the interaction of EWS with the spliceosome-associated YB-1 protein, resulting in the skipping of several exons in the MDM2 transcript. The resulting isoforms are unable to interact with P53; thus, P53 is stabilized and induces the expression of downstream targets involved in apoptosis and cell cycle arrest.

**Figure 4 biomolecules-11-01475-f004:**
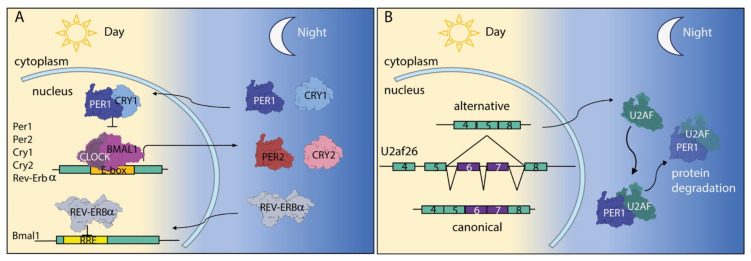
The circadian clock and RNA processing regulation. (**A**) Schematic representation of the circadian clock pathway. Upon circadian stimulation, the two transcription factors CLOCK and BMAL1 form a complex that binds the E-box motif in the promoter region of their target genes (such as PER1/2 CRY1/2 and Rev-Erbα) inducing their transcription. PER1 and CRY1 proteins then translocate into the nucleus to form a complex and repress CLOCK:BMAL1, thus forming a negative feedback mechanism that controls the pathway. In parallel, REV-Erbα translocates into the nucleus and contributes to repression of the transcription of BMAL1. (**B**) The U2AF26 transcript is alternatively spliced to yield two different isoforms, whose expression follows the circadian rhythms: the canonical isoform that includes exon 6 and 7 (purple exons) and the alternative one skipping them. The alternative isoform leads the formation of a different domain at the C-terminus, resulting in a new interaction of this splicing regulator with PER1 and promoting its degradation.

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
