# Peer review of "Coordination of RNA Processing Regulation by Signal Transduction Pathways"

_biomolecules, 2021, doi:10.3390/biom11101475_

Round 1

Reviewer 1 Report

The manuscript entitled “Coordination of RNA processing regulation by signal transduction pathways” by Ruta et al. reviews the mechanisms through which RNA processing events are post-translationally regulated by signaling pathways. This review is very thorough and is suitable for publication in its present form. I only have some minor grammatical/formatting suggestions.

Minor points:

  • Line 73: “ribonucleoproteins” should not have an “s” at the end
  • Line 81: “recognize” should be “recognizes”
  • Figure 1 title: “Splicing” does not need to be capitalized
  • Line 137: “element” should be “elements”
  • Line 174: remove “on”
  • Line 179: remove the “n” from “metazoan”
  • Line 195: “substrate” should be “substrates”
  • Line 204 and 205: italicize “in vitro”
  • Lines 265-268: these do not need to be spaced as bullet points and can instead be incorporated into a sentence
  • Line 274: “they” should be “the”
  • Line 297: italicize Arabidopsis thaliana and Drosophila melanogaster
  • Line 280: italicize saccharomyces cerevisiae
  • Line 324: italicize “in vitro”
  • Line 366 “suggest” should be “suggests”
  • Line 391: use the abbreviation for RNA binding proteins (RBPs)
  • Line 414: “member” should be “members”
  • Line 466: use the abbreviation for RNA binding proteins (RBPs)
  • Line 487: H2O2 should have the 2s as subscript
  • Line 653: remove “on”
  • Lines 667-670: a paragraph must be at least 3 sentences in length, so please expand on this sentence.
  • Line 701: “such positive feedback loop” should read “such a positive feedback loop”
  • Line 713: “promote” should be “promotes”
  • Lines 756-757: “can contribute to develop” should read “can contribute to the development”
  • Line 759: remove “by” before “HSP90”
  • Line 771: “nuclear stress bodies” should be “nuclear stress bodies”
  • Line 794: remove “on”
  • Line 795-796: italicize “Arabidopsis thaliana”
  • Line 796: there should not be such a big space and “-“ before “proteobacteria”
  • Line 813: italicize “Arabidopsis thaliana”
  • Line 892: “a similar regulation” should read “a similar mechanism”
  • Line 904: “transcriptsthat” is missing a space
  • Line 955: “post-translational mechanisms” should be “PTMs”
  • Line 981: italicize “Drosophila”
  • Line 1021: italicize “Arabidopsis thaliana”
  • Line 1044: “at C-terminal” should read “at the C-terminus”
  • Line 1046: “as tool” should read “as tools”
  • Line 1120: “tumour” should be “tumor” to be consistent with spelling throughout the document
  • Line 1159: “next” should be “near”
  • Throughout: avoid using unnecessary abbreviations that you only use once like “NMD”, “FTD”, “EMT” (not an exhaustive list, check all abbreviations)
  • Throughout: inconsistent use of commas before the word “and” in a list (Oxford comma). Sometimes a comma is used, and sometimes it is not – both are correct but you should be consistent.

Author Response

We thank the Reviewer for the positive comments to our manuscript and for pointing out the grammatical and formatting issues in the text. We have now followed her/his suggestion and introduced all changes using the “revision modality” of Word.

Reviewer 2 Report

The review by Ruta et al is an extremely comprehensive evaluation of the literature covering the role of RNA-binding proteins in coordinating nuclear RNA processing. The review begins by describing key elements of nuclear RNA processing then focuses on post-translational modifications that regulated RBP function. It then discusses the major signalling pathways and how they influence RNA processing and concludes with a more speculative section examining the possibility of sRNA-based therapeutic. The figures are generally clear, but they do need to be presented more clearly- the text labelling figure elements are too small to read easily.  For a review of this complexity, it would be good to have visually more appealing figures.

Overall, this review was well written and comprehensive review that will be of benefit to the scientific community.

Minor comments:

Line 378:  “RBPs recognize sequence-specific cis-acting elements…”. This is an oversimplification, as many RBP have little to no sequence specificity and either structural element. This sentence should be modified accordingly.

Lines 279-80: The species name need to be italicized.

Line 1159:  “However, in the next future it would be desirable ..”  delete “next”.

Author Response

We thank the Reviewer for the positive comments to our manuscript. As suggested, we have now modified the Figures to improve their readability and comprehension.

Minor comments:

Line 378:  “RBPs recognize sequence-specific cis-acting elements…”. This is an oversimplification, as many RBP have little to no sequence specificity and either structural element. This sentence should be modified accordingly.

As suggested by the Reviewer, we have now omitted this concept (see new line 408). The sequence-specific effect is now only referred to splicing factors that bind to enhancer or silencer elements in exons and introns (see new lines 153-154).

Lines 279-80: The species name need to be italicized. (new lines 300-302)

Line 1159: “However, in the next future it would be desirable ..”  delete “next”. (new line 1244)

We have changed the text as indicated by the Reviewer using the “revision modality” of Word. Please, note that line number is changed due to reformatting of the Figures.

Reviewer 3 Report

In this review by Ruta et al. authors provided the comprehensive overview of the signal transduction pathways and RNA metabolism links, particularly focussing on the regulation of splicing and poly-adenylation events by signalling pathways and how various kinds of post-translational modifications play a critical role in regulating these events. And finally discussed the potential opportunities of RNA based therapeutics by exploiting these signalling pathways.

Comments:

Fonts used in all the figure labels are very small, please increase the font size..

Check for minor typos..

For eg> line 668..pots to be corrected as post

Author Response

As suggested, we have now modified the Figures to improve their readability and comprehension.

We have now performed a spell check and corrected the grammatical and formatting issues indicated by the all four Reviewers. Changes in the text were introduced using the “revision modality” of Word.

Reviewer 4 Report

This is a thorough and well-written review on how various signal transduction pathways influence and control RNA processing, especially splicing.  It should be a helpful addition to members of the field or students wanting a ‘jump-start’ on this large body of work.  This is not my exact specialty so I learned a tremendous amount.

I have only a couple of suggestions and then a list of typos that need correcting.

  1. Because of the field, there are dozens of acronyms and ‘jargon’ which make it very difficult for the more naïve reader. I suggest that the authors collect these in a glossary at the end of the review.
  2. Line 377 – It would be helpful to put these sentences defining RBPs earlier when first mentioned.

Typos:

Line 80   should read kilodaltons

Lines 88, 104 and Fig. 1 – legend and text refer to active B complex while figure says complex C

Line 114 should read promotes

Line 189 should read grades

Line 195 should read substrates

Line 497 – remove one of the ‘phosphorylation’

Line 686 should read ‘pathway that relies’

Line 713 should read promotes

Line 743 should read an rather than and

Line 796. Missing Greek symbol

Line 812  should read genes was regulated

Line 870 should read This PTM…

Line 904.  Need space between ‘transcriptsthat’

Line 952 should read machinery that exist among

Line 965 should read stability of the

Author Response

As suggested, we have now added a list of all abbreviations used in the text. The List is at pages 25-26 of the revised manuscript.

As suggested, we have now moved the description earlier (see new line 153-154). We have also performed a spell check and corrected the grammatical and formatting issues indicated by the Reviewer. Changes in the text were introduced using the “revision modality” of Word.